# Molecular Networks of Platinum Drugs and Their Interaction with microRNAs in Cancer

**DOI:** 10.3390/genes14112073

**Published:** 2023-11-13

**Authors:** Shihori Tanabe, Eger Boonstra, Taehun Hong, Sabina Quader, Ryuichi Ono, Horacio Cabral, Kazuhiko Aoyagi, Hiroshi Yokozaki, Edward J. Perkins, Hiroki Sasaki

**Affiliations:** 1Division of Risk Assessment, Center for Biological Safety and Research, National Institute of Health Sciences, Kawasaki 210-9501, Japan; 2Department of Bioengineering, Graduate School of Engineering, The University of Tokyo, Tokyo 113-0033, Japanhongtaehun1125@g.ecc.u-tokyo.ac.jp (T.H.); horacio@bmw.t.u-tokyo.ac.jp (H.C.); 3Innovation Centre of NanoMedicine (iCONM), Kawasaki Institute of Industrial Promotion, Kawasaki 210-0821, Japan; sabina-q@kawasaki-net.ne.jp; 4Division of Cellular and Molecular Toxicology, Center for Biological Safety and Research, National Institute of Health Sciences, Kawasaki 210-9501, Japan; onoryu@nihs.go.jp; 5Department of Clinical Genomics, National Cancer Center Research Institute, Tokyo 104-0045, Japan; kaaoyagi@ncc.go.jp; 6Department of Pathology, Kobe University of Graduate School of Medicine, Kobe 650-0017, Japan; hyoko@med.kobe-u.ac.jp; 7US Army Engineer Research and Development Center, Vicksburg, MS 39180, USA; edward.j.perkins@erdc.dren.mil; 8Department of Translational Oncology, National Cancer Center Research Institute, Tokyo 104-0045, Japan; hksasaki@ncc.go.jp

**Keywords:** cisplatin, drug resistance, microRNA, molecular network, pathway analysis, platinum drug, tumor microenvironment

## Abstract

The precise mechanism of resistance to anti-cancer drugs such as platinum drugs is not fully revealed. To reveal the mechanism of drug resistance, the molecular networks of anti-cancer drugs such as cisplatin, carboplatin, oxaliplatin, and arsenic trioxide were analyzed in several types of cancers. Since diffuse-type stomach adenocarcinoma, which has epithelial–mesenchymal transition (EMT)-like characteristics, is more malignant than intestinal-type stomach adenocarcinoma, the gene expression and molecular networks in diffuse- and intestinal-type stomach adenocarcinomas were analyzed. Analysis of carboplatin revealed the causal network in diffuse large B-cell lymphoma. The upstream regulators of the molecular networks of cisplatin-treated lung adenocarcinoma included the anti-cancer drug trichostatin A (TSA), a histone deacetylase inhibitor. The upstream regulator analysis of cisplatin revealed an increase in FAS, BTG2, SESN1, and CDKN1A, and the involvement of the tumor microenvironment pathway. The molecular networks were predicted to interact with several microRNAs, which may contribute to the identification of new drug targets for drug-resistant cancer. Analysis of oxaliplatin, a platinum drug, revealed that the SPINK1 pancreatic cancer pathway is inactivated in ischemic cardiomyopathy. The study showed the importance of the molecular networks of anti-cancer drugs and tumor microenvironment in the treatment of cancer resistant to anti-cancer drugs.

## 1. Introduction

Metallodrugs, including platinum drugs and arsenic trioxide, are widely used as cancer therapeutics [1,2,3,4,5]. Platinum drugs, such as cisplatin and carboplatin, are commonly used for the treatment of a variety of cancers, including bladder, head and neck, lung, gastrointestinal, ovarian, and testicular cancer [6,7]. Arsenic trioxide is also the front-line drug for the treatment of acute promyelocytic leukemia [8,9]. Arsenic trioxide has a long history of medicinal use for thousands of years, despite its toxic effects as a poison [10,11]. Arsenic trioxide is effective for patients with recurrent acute promyelocytic leukemia [12]. An increasing number of patients have been treated with arsenic trioxide; however, the precise mechanism of action is not fully understood [10,11].

The principal mechanism of cisplatin-induced cytotoxicity is the formation of platinum-DNA adducts, for which cellular reactions attenuating the DNA damage are key factors in resistance to cisplatin [13]. Cisplatin treatment causes several toxic side effects, including hepatotoxicity, cardiotoxicity, and nephrotoxicity, which may be based on the nature of cisplatin interacting with DNA and forming covalent adducts with purine DNA bases [6]. The mechanisms of cisplatin resistance include pre-target resistance relating to the reduction of cisplatin uptake, the increase in cisplatin effusion to the extracellular space or the increased sequestration of cisplatin by glutathione, on-target resistance involving molecular damage caused by DNA binding of cisplatin, post-target resistance, including signaling pathways leading to cell death triggered by adducts, and off-target resistance relating to alterations in signaling pathways that interfere with cisplatin-induced proapoptotic events [14,15]. Furthermore, epigenetic mechanisms regulating gene expression in non-coding RNA basis and metabolic processes, including glycometabolism and amino acid and lipid metabolism, are associated with resistance to cisplatin treatment [14,15]. In terms of uptake and efflux of platinum drugs, polymorphisms of transporters influence platinum drugs [7]. Although numerous research studies have examined the mechanism of chemoresistance, the extent to which metallodrugs produce drug resistance has not yet been fully revealed [16,17,18,19].

A study in oral squamous cell carcinoma suggested that microRNA (miRNA)-485-5p targets keratin 17 to regulate oral cancer stemness and chemoresistance through integrin, protein tyrosine kinase 2 (PTK2 or FAK), Src, mitogen-activated protein kinase 1 (MAPK1 or ERK), and the β-catenin pathway [16]. *N*^6^-methyladenosine RNA methylation of centromere protein K in cervical cancer promoted stemness and chemoresistance. The activation of Wnt/β-catenin signaling leads to the enhancement of DNA damage repair pathways that are necessary for cisplatin/carboplatin resistance and the epithelial–mesenchymal transition (EMT) that is involved in metastasis [17].

In non-small cell lung cancer, miRNAs play roles in the resistance to platinum-based chemotherapy such as adjuvant cisplatin (CDDP, *cis*-diammine-dichloro-platinum II) [18]. It has been suggested that miR-129-1-3p, miR-155, miR-200c, the miR-17 family (-17, 20a, 20b), miR-15b, miR-27, and miR-181a are involved in CDDP resistance and the EMT in non-small cell lung cancer [18]. Recent studies demonstrated that mRNA, miRNA, cirRNA, and metabolites were related to the resistance of cisplatin in cancer [20,21,22].

It has been reported that miRNAs are involved in cisplatin response by regulating the EMT [23]. The EMT, a cellular phenomenon in which cell types change from epithelial type to mesenchymal type, is involved in the malignancy of cancers, including proliferation, invasion, and metastasis [24]. The EMT phenotype is regulated by miRNAs and plays important roles in drug resistance that is implicated with cancer stemness [24]. Tumor-derived exosomes containing several miRNAs promote the EMT and cancer-cell invasion [25]. Although the EMT is involved in several signaling pathways, such as the phosphatase and tensin homolog (PTEN)/phosphatidylinositol 3-kinase (PI3K), Wnt/β-catenin, and TGF-β pathways, the precise correlation among the EMT, exosomes, and drug resistance is unknown [25]. The EMT is one of the features of cancer malignancy and treatment resistance [26,27,28,29,30]. Previous studies demonstrated that the EMT is implicated with cisplatin resistance [31,32,33]; however, the precise mechanisms and the underlying networks are not fully understood. In this context, the present study investigated the relationship between metallodrugs and the EMT and tumor microenvironment pathways. Additionally, the molecular networks of carboplatin, cisplatin, and arsenic trioxide were investigated in the study.

## 2. Materials and Methods

### 2.1. RNA Sequencing Data Collection

The RNA sequencing data of diffuse- and intestinal-type stomach adenocarcinomas are publicly available in The Cancer Genome Atlas (TCGA) of the cBioPortal for Cancer Genomics database at the National Cancer Institute (NCI) Genomic Data Commons (GDC) data portal [34,35,36,37]. Publicly available data on stomach adenocarcinoma in the TCGA, Stomach Adenocarcinoma (TCGA, PanCancer Atlas) [34,36] were compared between diffuse-type stomach adenocarcinoma, which is genomically stable (n = 50), and intestinal- stomach adenocarcinoma, which has a feature of chromosomal instability (n = 223), in TCGA Research Network publications, as previously described [34,38,39].

### 2.2. Network Pathway Analysis

Data on intestinal- and diffuse-type stomach adenocarcinomas from the TCGA cBioPortal for Cancer Genomics were uploaded and analyzed through the use of Ingenuity Pathway Analysis (IPA) (QIAGEN Inc., Hilden, Germany) [40]. In the IPA database, causal networks were analyzed and bioprofiler analysis was performed on carboplatin. Causal networks, canonical pathways, and regulatory networks were analyzed in gastric adenocarcinoma data, lung adenocarcinoma data, diffuse large B-cell lymphoma data, and arsenic trioxide treatment data in the IPA database.

### 2.3. Data Visualization

The results of network analyses of causal networks of cisplatin treatment were visualized with Tableau 2023.3 software (https://www.tableau.com (accessed on 7 November 2023)).

### 2.4. Statistic Analysis

The RNA sequencing data on diffuse- and intestinal-type stomach adenocarcinomas were analyzed via Student’s *t*-test. The z-scores of intestinal- and diffuse-type stomach adenocarcinoma samples were compared, and the difference was considered significant at *p* < 0.00001, following previous reports [38,41]. The activation z-score in each pathway was calculated in IPA to show the level of activation.

## 3. Results

### 3.1. Causal Networks of Carboplatin Activity Plot in Cancer Treatment

Platinum drugs were searched with the term “platinum” in Ingenuity Pathway Analysis (IPA), which included carboplatin and cisplatin (Table 1). To reveal the causal network of carboplatin treatment, master regulators in depth 3 were investigated in the carboplatin activity plot in IPA analysis. The analysis in the top z-score of 790 analyses for carboplatin in depth 3 of the master regulators was in diffuse large B-cell lymphoma (DLBCL) (as of 2022) (Figure 1a). In this analysis, N-(furan-2-ylmethyl)-8-(4-methylsulfonylphenyl)-[1,2,4]triazolo[4,3-c]pyrimidin-5-amine (EED226) (5 µM, 24 h), a potent and selective inhibitor of polycomb repressive complex 2 (PRC2) that directly binds to the histone H3 lysine 27 binding pocket of embryonic ectoderm development (EED), a subunit of PRC2, was treated to DLBCL [42]. Bioprofiler analysis on carboplatin identified the relationship between carboplatin and anaplastic lymphoma kinase receptor tyrosine kinase (ALK) mutation-negative CD274 positive non-small cell lung cancer, ALK mutation negative epidermal growth factor receptor (EGFR) sensitizing mutation negative non-squamous non-small cell lung cancer, CD274 low expression positive non-squamous non-small cell lung cancer, EGFR mutation-negative CD274 positive non-small cell lung cancer, or locally advanced non-squamous non-small cell lung cancer (Figure 1b,c). Canonical pathways regulation of the EMT by the growth factors pathway, the tumor microenvironment pathway, the nuclear factor erythroid 2-related factor 2 (NRF2)-mediated oxidative stress response, the regulation of the EMT in the development pathway, and the regulation of the EMT pathway were related to the carboplatin network (Figure 1b,c). Drugs were identified to have a direct relationship between the network, which included clazakizumab, and anti-interleukin (IL)-6 monoclonal antibody (Figure 1b,c). The IL-6 pathway was one of the components of signaling pathways in the regulation of the EMT by the growth factors pathway (Figure 1d). IL-6 activates the IL6 receptor (IL6R), leading to activation of Janus kinase (JAK), signal transducer and activator of transcription 3 (STAT3) and twist family bHLH transcription factor 1 (TWIST1), which subsequently induces cell migration, cell invasion, and metastasis (Figure 1d). Molecules in the regulation of the EMT by the growth factors pathway are listed in Table 2.

### 3.2. Causal Networks of Cisplatin-Treated Lung Adenocarcinoma

Causal networks of cisplatin treatment were investigated in IPA. In the IPA database, 169 analyses and 171 datasets were found to be related to cisplatin, among which 12 analyses were identified as having treatment of cisplatin (as of 2022) (Table 3). The numbers in the “Analysis Name” column in Table 3 indicate the ID name of the Ingenuity Pathway Analysis database (As of 2022). Causal network analysis of the cisplatin-treated samples revealed camptothecin as a master regulator of the causal network (Figure 2a). The causal network of camptothecin in cisplatin-treated lung adenocarcinoma included Fas cell surface death receptor (FAS) in the tumor microenvironment pathway, phosphatase and tensin homolog (PTEN) and MDM2 proto-oncogene (MDM2) in the cancer drug resistance by drug efflux and interferon regulatory factor 1 (IRF1) in the production of nitric oxide and reactive oxygen species in macrophages (Figure 2b). Upstream regulator analysis of cisplatin-treated lung adenocarcinoma revealed the upregulation in FAS, protein kinase C α (PRKCA), and cyclin-dependent kinase inhibitor 1A (CDKN1A) in the molecular mechanisms of cancer, and the involvement of the tumor microenvironment pathway in the causal network of cisplatin (Figure 3). CDKN1A, MDM2, and PTEN, which are involved in glioblastoma multiforme signaling, and PRKCA, which is involved in the production of nitric oxide and reactive oxygen species in macrophages, were also included in the cisplatin network (Figure 3).

### 3.3. Cisplatin as an Upstream Regulator of Gastric Adenocarcinoma

The upstream regulator analysis of cisplatin in the activity plot revealed 64,880 analyses for cisplatin (Figure 4a). Cisplatin was identified as an activated upstream regulator in pembrolizumab, a humanized IgG4 monoclonal antibody against programmed cell death-1 (PD-1)-treated gastric adenocarcinoma in progressive disease (Figure 4b) [43]. The mechanistic network of cisplatin included epidermal growth factor receptor (EGFR) and SMAD family member 3 (SMAD3) in the regulation of the EMT by the growth factors pathway and Akt, activator protein 1 (Ap1), early growth response 1 (EGR1), Fos proto-oncogene (FOS), Jun proto-oncogene (JUN), NFκB (complex), RELA proto-oncogene (RELA), and signal transducer and activator of transcription 3 (STAT3) in the regulation of the EMT by the growth factors pathway and the tumor microenvironment pathway (Figure 4b). miR-1195 (miRNAs w/seed GAGUUCG), mir-124, mir-338, mir-379, mir-434, mir-515, mir-622, MIR124, MIR499B, and CpG ODN/STAT3 siRNA CAS3/SS3 (CAS3/SS3) were identified as microRNAs and siRNA to have direct interactions with the network of cisplatin (Figure 4b, Table 4). Hypoxia-inducible factor 1 subunit α (HIF1A), STAT3, and RELA were the target molecules of the miRNAs. The tumor microenvironment pathway was activated in the pembrolizumab-treated gastric adenocarcinoma (Figure 4c). IL-6 and SPP1, which are regulated by M2 tumor-associated macrophages, were upregulated in the network (Figure 4c). Since the causal networks of camptothecin were activated in cisplatin-treated lung adenocarcinoma, the gene expression profiles of diffuse- and intestinal-type stomach adenocarcinomas were investigated in the causal network of camptothecin. Camptothecin was activated in intestinal-type stomach adenocarcinoma and inactivated in diffuse-type stomach adenocarcinoma (Figure 5).

### 3.4. Arsenic Trioxide Treatment in the EMT by Growth Factors Pathway

#### 3.4.1. Regulatory Networks in Arsenic-Trioxide-Treated Liver Carcinoma

The network of arsenic trioxide was investigated in the EMT by the growth factors pathway, which identified regulatory networks of arsenic-trioxide-treated liver carcinoma (Table 5) [44]. The top regulatory network was involved in the cell death of carcinoma cell lines and cell proliferation of adenocarcinoma cell lines. Regulators of the network were amphiregulin (AREG), cytoskeleton-associated protein 2 like (CKAP2L), H2A.Z variant histone 1 (H2AZ1), HNF1A antisense RNA 1 (HNF1A-AS1), immunity-related GTPase family M member 1 (Irgm1), lin-9 DREAM MuvB core complex component (LIN9), MYB proto-oncogene like 2 (MYBL2), PCNA clamp associated factor (PCLAF), and S100 calcium binding protein A6 (S100A6). AREG is an epidermal growth factor receptor (EGFR) ligand located in extracellular space.

#### 3.4.2. Causal Networks of Arsenic and Direct Interaction with microRNAs

A causal network of arsenic was activated in intestinal-type stomach adenocarcinoma and inactivated in diffuse-type stomach adenocarcinoma (Figure 6). The causal network of arsenic had direct RNA–RNA interactions with mir-101, mir-103, miR-125b-2-3pp (and other miRNAs w/seed CAAGUCA), mir-145, mir-22, mir-338, mir-379, mir-515, mir-622, and MIR499B (Table 6). mir-101 targets MTOR and mir-22 targets SP1, mir-103, mir-145, and mir125b-2-3pp (and other miRNAs w/seed CAAGUCA) target TP53, which are inhibited by arsenic (Figure 6). mir-370, mir-515, mir-338, MIR499B and mir-622 targets HIF1A, which is activated by arsenic (Figure 6).

### 3.5. Analysis of Oxaliplatin and SPINK1 Pancreatic Cancer Pathway

Network analysis of oxaliplatin revealed that oxaliplatin as a master regulator was predicted to be inhibited, and the serine peptidase inhibitor Kazal type 1 (SPINK1) pancreatic cancer pathway was inactivated in ischemic cardiomyopathy (Figure 7). EGFR involved in the SPINK1 general cancer pathway, regulation of the EMT by the growth factors pathway, the pulmonary fibrosis idiopathic signaling pathway, and hepatic fibrosis/hepatic stellate cell activation were decreased in the oxaliplatin-inhibitory network (Figure 7a). On the other hand, B cell leukemia/lymphoma 2 apoptosis regulator (BCL2), which is involved in the pulmonary fibrosis idiopathic signaling pathway, hepatic fibrosis/hepatic stellate cell activation, the coronavirus pathogenesis pathway, the tumor microenvironment pathway, and autophagy were increased in the oxaliplatin-inhibitory network (Figure 7a). The SPINK1 pancreatic cancer pathway consists of two main streams of signaling, which are the coagulation factor II (thrombin) receptor-like 1 (F2RL1)-transforming growth factor β receptor (TGFBR)-Smad2/3-metastasis axis and the serine protease (PRSS)–injury of pancreatic cells–acute recurrent pancreatitis–chronic pancreatitis axis (Figure 7b).

## 4. Discussion

Causal network analysis was used to investigate the molecular networks in treatment-resistant cancer, and the results revealed that EED226, a potent inhibitor of PRC2, is involved in DLBCL [42]. EED226 inhibits PRC2, which is resistant to an inhibitor of enhancer of zeste 2 polycomb repressive complex 2 subunit (EZH2), a subunit of PRC2 [42]. Since EED226 targets the trimethylated H3K27 binding pocket of EED, histone modification may be involved in cancer-treatment resistance.

Several miRNAs, MIR155, MIR192, MIR34A, and MIR200, are components of the regulation of the EMT by the growth factors pathway. mir-155_5p was upregulated at lung metastasis and peritoneal metastasis of colorectal cancer, compared to primary colorectal cancer [45]. Prior investigation of relationships between platinum drugs such as cisplatin and carboplatin and non-coding RNAs demonstrated that miRNAs are correlated with cisplatin activity [46]. Additionally, recent studies revealed the association between the EMT and miRNAs and their roles in drug resistance [24,47,48,49]. Downregulation of miRNA-214 induces the EMT and the migration and invasion of gastric cancer [50].

Pembrolizumab, a humanized IgG4 monoclonal antibody against PD-1, is an immune checkpoint inhibitor and can be used for microsatellite instability-high or tumor mutational burden-high advanced gastric cancer [51]. The overall response rate of pembrolizumab in advanced PD-L1-positive gastric cancer was 22.2% [52]. The patients in the IL-1R1^high^ subgroup demonstrated a significantly lower response to pembrolizumab than those in the IL-1R1^low^ subgroup [53]. Sundar et al. found a potential role of alternative promoter utilization as a predictive biomarker for resistance to immune checkpoint inhibition [54]. The interferon (IFN)-γ-related gene-expression profile predicted response to pembrolizumab [55]. Since the tumor microenvironment pathway was activated in pembrolizumab-treated gastric adenocarcinoma, the treatment resistance is involved in the tumor microenvironment pathway.

In the current study, HIF1A was identified as a target of several miRNAs in the causal network of arsenic. HIF1A and CDKN1A, together with miR-3607-3p, miR-301a-3p, and miR-93-5p, are associated with prolonged survival in glioblastoma patients treated with regorafenib [56]. miR-101 induces HIF1A-mediated apoptosis and cell cycle arrest [57]. HIF1A may be regulated by miRNAs in the causal network of arsenic. Since the activity state of the EMT pathway can be predicted in artificial-intelligence-based modeling, molecular network pathway analysis may be applicable to predict the molecular-induced responses [41].

F2RL1, F2R like trypsin receptor 1 or proteinase-activated receptor 2 (PAR2), was identified in the SPINK1 pancreatic cancer pathway. F2RL1 is a G protein-coupled receptor that is activated by distinct serine proteases to regulate receptor-specific signaling pathways involved in cell migration [58]. PAR2/PAR1 and TGF-β/ transforming growth factor β receptor 1 (TGFBR1 or ALK5) form a complex regulatory network in fibrosis and cancer [59]. PAR2, highly expressed in pancreatic ductal adenocarcinoma, is essential for TGF-β1-induced cell motility and TGFBR1 expression [60]. Future investigation is required to explore the RNA regulation of F2RL1 and the signaling pathways.

Regarding RNA regulation, miRNAs, including mir-124, mir-338, mir-379, mir-434, mir-515, and mir-622, were identified as interacting with the cisplatin network in gastric adenocarcinoma. mir-124 is implicated in regulating the sensitivity of renal cancer cells to cisplatin-induced necroptosis [61]. The sensitivity of lung cancer cells to cisplatin is changed by mir-124 [62]. These recent reports suggested that mir-124 oriented regulation of sensitivity of cancer toward cisplatin. The molecular networks were predicted to interact with several miRNAs, which may contribute to the identification of new drug targets for drug-resistant cancer.

The main mechanism of drug resistance is a DNA-platinum drug complex, which induces a DNA repair system [7]. The limitation of this study is the lack of elucidating the exact mechanism regarding the correlation between platinum complexes and the EMT. Platinum drug treatment could coactivate DNA repair mechanisms, cell cycles, and the EMT signaling pathways. It is also possible that some EMT regulation is involved during platinum drug treatment. Clarifying these possibilities would be a future challenge.

In this study, we investigated the altered molecular networks in metallodrugs, including platinum drugs and arsenic trioxide, and their relation to miRNAs, which will contribute to revealing the precise mechanism of drug resistance and possible combination therapy in clinical applications. The identification of target miRNAs in molecular networks would lead to the development of cancer therapy and strategies for precision medicines in practice.

## 5. Conclusions

Platinum drug treatment is related to the tumor microenvironment pathway in cancer. Cisplatin was identified as the upstream regulator of pembrolizumab-treated gastric adenocarcinoma. Several microRNAs were identified to interact with the cisplatin network. The carboplatin network was related to the regulation of the EMT pathway and the tumor microenvironment pathway. A close correlation between anti-cancer drug resistance and tumor microenvironment pathways needs to be revealed in future investigations.

## Figures and Tables

**Figure 1 genes-14-02073-f001:**
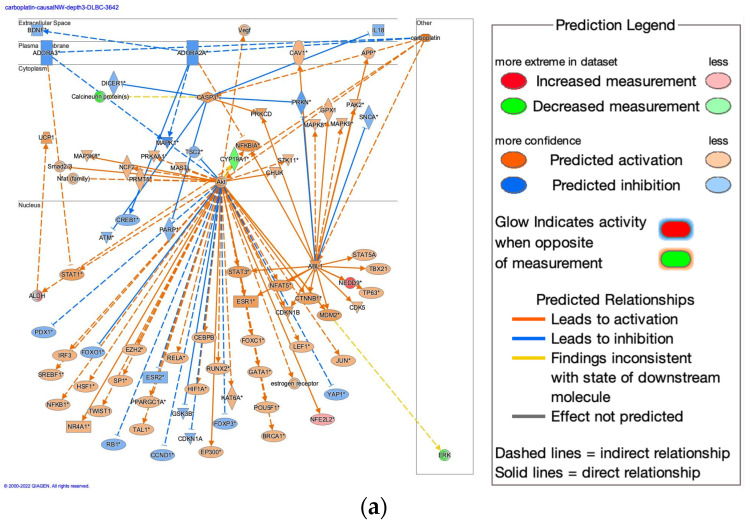
Carboplatin molecular network in cancer. (**a**) Causal network of carboplatin in EED226-treated diffuse large B-cell lymphoma. Gene identifiers marked with an asterisk (*) indicate that multiple identifiers in the dataset file map to a single gene in the Global Molecular Network. (**b**) Carboplatin bioprofiler network in diffuse-type stomach adenocarcinoma. (**c**) Carboplatin bioprofiler network in intestinal-type stomach adenocarcinoma. (**d**) Regulation of the EMT by growth factors pathway colored with expected activation states. The red color indicates activation, while the green color indicates inactivation.

**Figure 2 genes-14-02073-f002:**
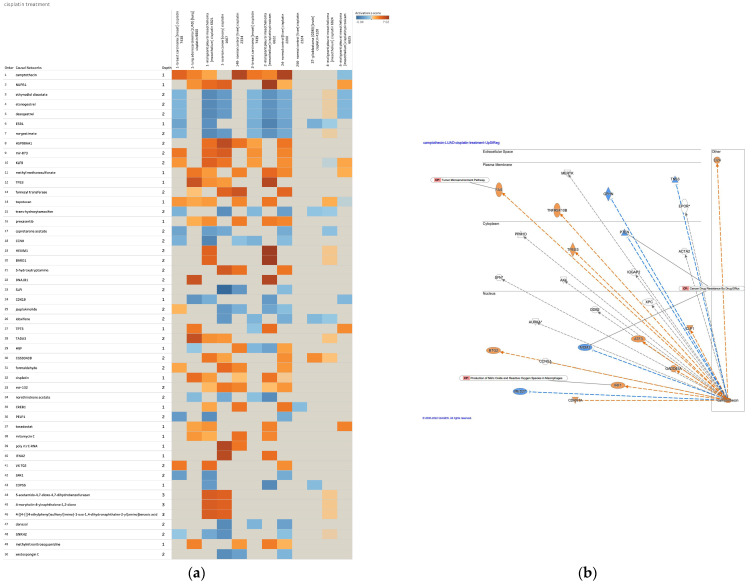
Causal networks in cisplatin treatment and involvement of tumor microenvironment pathway. (**a**) Camptothecin was identified as a master regulator in cisplatin-treated samples. The activity z-score of the top 50 master regulators is shown in the heat map. (**b**) Tumor microenvironment pathway-related Fas cell surface death receptor (FAS) was located in the plasma membrane in the network of camptothecin in cisplatin-treated lung adenocarcinoma. Gene identifiers marked with an asterisk (*) indicate that multiple identifiers in the dataset file map to a single gene in the Global Molecular Network.

**Figure 3 genes-14-02073-f003:**
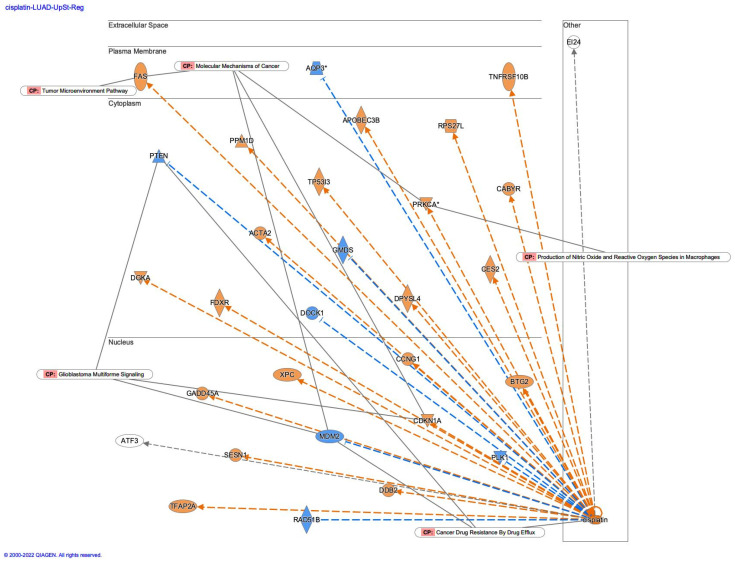
Upstream regulator analysis of cisplatin revealed the upregulation in FAS, PRKCA, and CDKN1A, and involvement of tumor microenvironment pathway (lung adenocarcinoma (LUAD) [lung] cisplatin 9800). Gene identifiers marked with an asterisk (*) indicate that multiple identifiers in the dataset file map to a single gene in the Global Molecular Network.

**Figure 4 genes-14-02073-f004:**
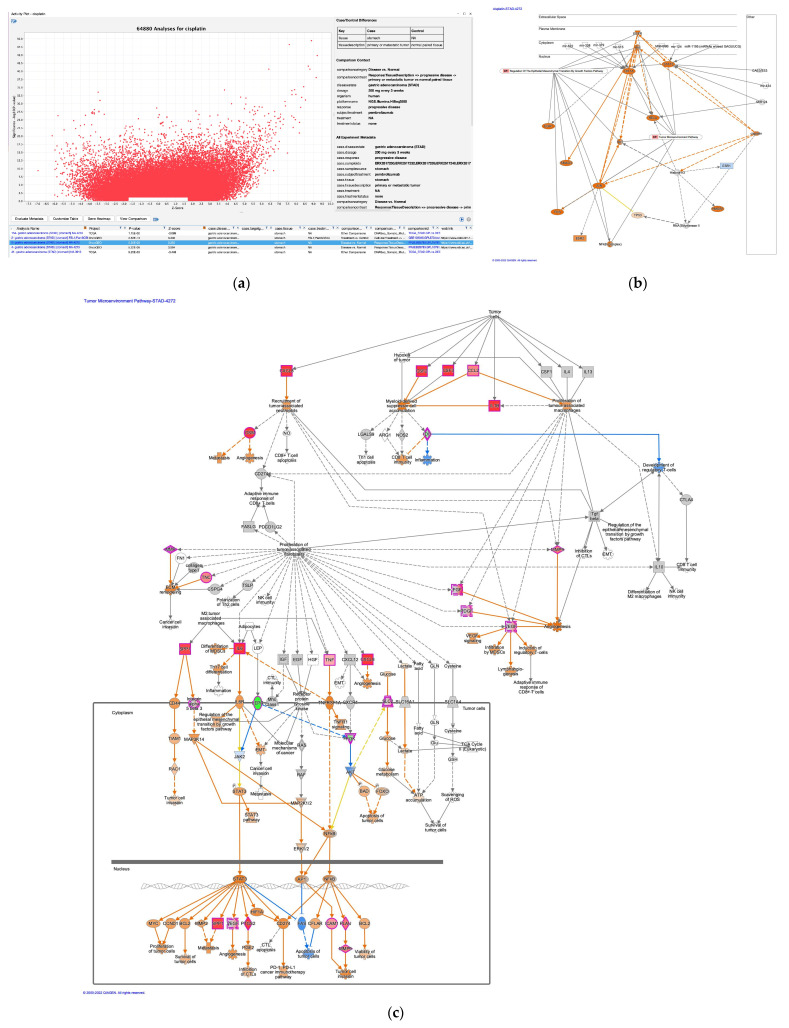
Cisplatin as an upstream regulator in gastric adenocarcinoma. (**a**) The upstream regulator analysis of cisplatin in the activity plot revealed 64,880 analyses for cisplatin. (**b**) Cisplatin was identified as an activated upstream regulator in pembrolizumab, a humanized IgG4 monoclonal antibody against programmed cell death-1 (PD-1)-treated gastric adenocarcinoma in progressive disease. (**c**) The tumor microenvironment pathway was activated in the pembrolizumab-treated gastric adenocarcinoma.

**Figure 5 genes-14-02073-f005:**
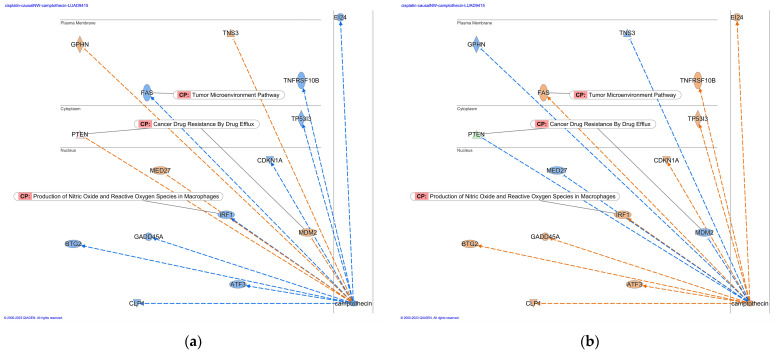
Causal network of camptothecin in diffuse- and intestinal-type stomach adenocarcinomas. (**a**) The network of camptothecin in diffuse-type stomach adenocarcinoma. (**b**) The network of camptothecin in intestinal-type stomach adenocarcinoma.

**Figure 6 genes-14-02073-f006:**
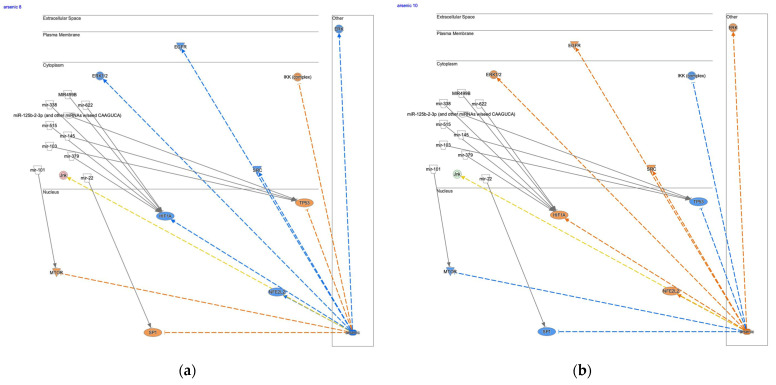
Causal network of arsenic in diffuse- and intestinal-type stomach adenocarcinomas. (**a**) The network of arsenic in diffuse-type stomach adenocarcinoma. (**b**) The network of arsenic in intestinal-type stomach adenocarcinoma.

**Figure 7 genes-14-02073-f007:**
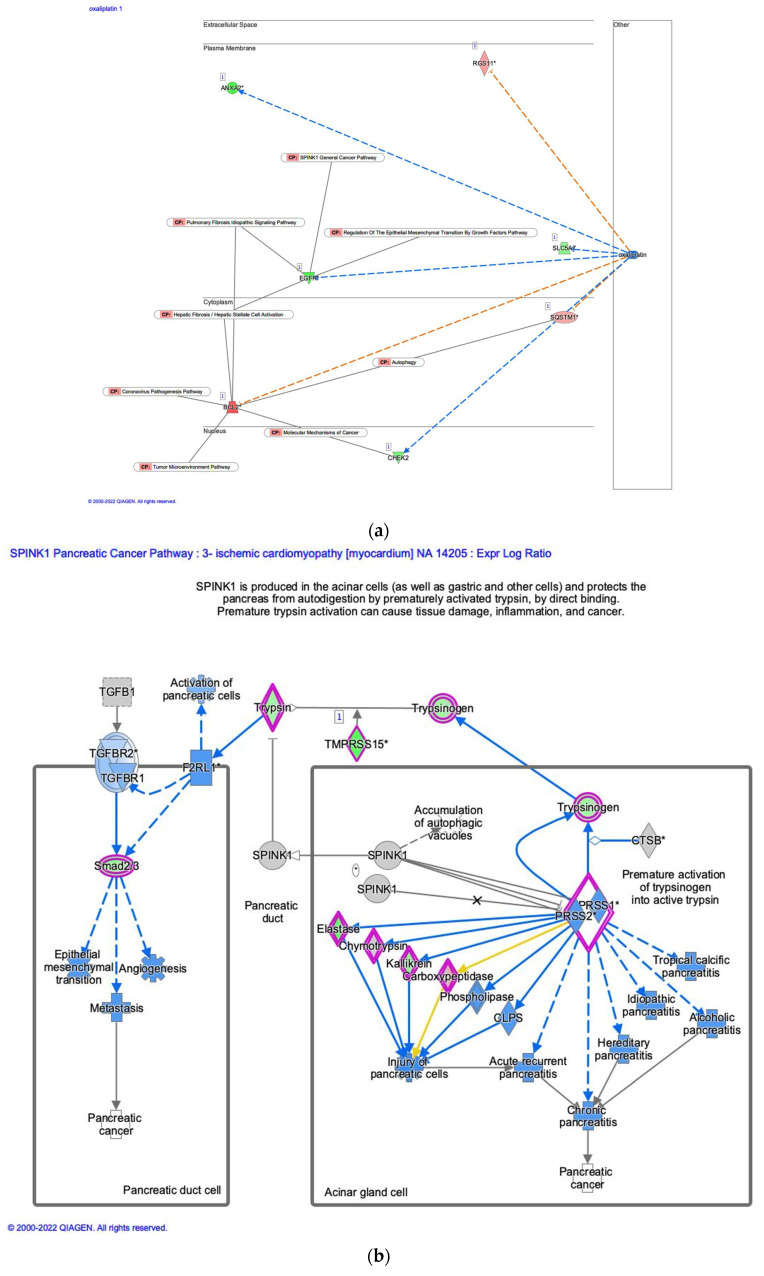
Network analysis of oxaliplatin revealed that oxaliplatin as a master regulator was predicted to be inhibited and the SPINK1 pancreatic cancer pathway was inactivated in ischemic cardiomyopathy. (**a**) The network of oxaliplatin. Oxaliplatin was predicted to be inhibited as a master regulator in ischemic cardiomyopathy. (**b**) The SPINK1 pancreatic cancer pathway was inactivated in ischemic cardiomyopathy. Gene identifiers marked with an asterisk (*) indicate that multiple identifiers in the dataset file map to a single gene in the Global Molecular Network.

**Table 1 genes-14-02073-t001:** Platinum drugs were searched with the term “platinum” in Ingenuity Pathway Analysis.

Number	Symbol
1	platinum
2	Pt^2+^
3	carboplatin
4	dicycloplatin
5	enloplatin
6	eptaplatin
7	iproplatin
8	nedaplatin
9	oxaliplatin
10	picoplatin
11	satraplatin
12	sebriplatin
13	zeniplatin
14	platinum agent
15	platinum chemotherapy regimen
16	platinum-based doublet chemotherapy
17	E platinum
18	platinum (II) chloride
19	platinum agent/trastuzumab
20	cisplatin
21	platinum chemotherapy regimen/radiotherapy
22	platinum-based doublet chemotherapy/taxane
23	platinum chemotherapy regimen/vinorelbine
24	platinum chemotherapy/taxoid derivative
25	platinum-based doublet chemotherapy/vinorelbine
26	platinum-based triplet chemotherapy
27	adjuvant chemotherapy/platinum agent
28	cetuximab/platinum chemotherapy regimen
29	gemcitabine/platinum chemotherapy
30	paclitaxel/platinum chemotherapy regimen
31	platinum-based neoadjuvant chemotherapy
32	bevacizumab/platinum chemotherapy/taxoid derivative
33	platinum-norsperimidine complex Pt3NSpd2
34	capecitabine/platinum chemotherapy regimen
35	bevacizumab/gemcitabine/platinum chemotherapy regimen
36	platinum chemotherapy regimen/thoracic radiotherapy
37	nonplatinum-based doublet chemotherapy
38	platinum acetylacetonate-titanium dioxide nanoparticles
39	BP-Cx1-platinum complex BP-C1
40	cetuximab/5-fluorouracil/platinum chemotherapy
41	etoposide/platinum chemotherapy regimen
42	ormaplatin
43	Bamet-UD2
44	NC-4016
45	fluoropyrimidine/platinum-based triplet chemotherapy
46	(diaminocyclohexane)(diacetato)(dichloro)platinum
47	lobaplatin
48	non-pemetrexed containing platinum chemotherapy regimen
49	triplatin tetranitrate
50	dacplatinum
51	cisplatin/etoposide
52	cisplatin/docetaxel
53	cisplatin/paclitaxel
54	carboplatin/paclitaxel
55	cisplatin/epirubicin/5-fluorouracil

**Table 2 genes-14-02073-t002:** Molecules involved in the EMT by the growth factors pathway (Ingenuity Pathway Analysis).

Location	Symbol	
Cytoplasm	AKT	MIR155
	CDC42	MIR192
	DOCK10	MIR34A
	ERK1/2	MKK3/6
	FRS2-GRB2-SHP2	p38MAPK
	GAB1	PI3K
	GRB2	PTPN11
	Grb2-Shc1-Sos	RAF
	Ikk	Ras
	JAK	RHOA
	Jnk	SHC1
	MAP2K4/7	SMAD2/3
	MAP3K1	SMURF1
	MAP3K7	SOS
	MAPK1	TAB1
	MEK	VIM
	MEST	
Extracellular Space	EGF	MMP2
	FGF	MMP9
	FGF dimer	PDGF
	HGF	Tgfbeta
	IL6	TGFB2
	MMP1	
Nucleus	AP1	MTOR
	EGR1	NFkappaB
	ESRP2	SMAD2/3/4
	ETS1	SMAD4
	FOS	SNAI1
	FOXC2	SNAI2
	FOXO1	STAT3
	GSC	TCF3
	GSK3B	TWIST1
	HMGA2	ZEB1
	HSF1	ZEB2
	ID2	
Plasma Membrane	CDH1	MET
	CDH2	OCLN
	CLDN3	Pdgfr
	EGFR	PDGFRA
	ERBB2	RAC1
	FGFR	TGFBR1
	FRS2	TGFBR2
	IL6R	Tnfreceptor
Other	LATS	PAR6
	MIR200	Tnf
	N-Cadherin	YAP/TAZ

**Table 3 genes-14-02073-t003:** Analyses of cisplatin treatment (Ingenuity Pathway Analysis).

Analysis Name	Comparison Contrast	Upregulated log2 Cutoff	Project Name	Organism
1- breast carcinoma [breast] cisplatin 7438	Sampling Time => 10 to 11 h after treatment vs. NA	0.344	GSE28274	human
1- lung adenocarcinoma (LUAD) [lung] cisplatin 9800	Treatment => cisplatin vs. none	0.1224	GSE6410	human
1- malignant pleural mesothelioma [mesothelium] cisplatin 6821	Treatment: Treat Time [hours] => 24 -> cisplatin vs. none	0.1633	GSE22445	human
1- ovarian cancer [ovary] cisplatin 1667	Experiment Group => cisplatin 1 day recovery 2 weeks vs. monolayer culture	0.6374	GSE144232	human
140- normal control [liver] cisplatin 2334	Treat Time [days]: Treatment => 1 -> cisplatin vs. DMSO	1.1422	GSE57805	rat
2- breast carcinoma [breast] cisplatin 7439	Sampling Time => 8 to 9 h after treatment vs. NA	0.0815	GSE28274	human
2- malignant pleural mesothelioma [mesothelium] cisplatin; piroxicam 6822	Treatment: Treat Time [h] => 24 -> cisplatin; piroxicam vs. none	0.539	GSE22445	human
24- normal control [liver] cisplatin 2200	Treat Time [days]: Treatment => 0.67 -> cisplatin vs. DMSO	1.0049	GSE57805	rat
256- normal control [liver] cisplatin 2324	Treatment: Treat Time [days] => cisplatin -> 1 vs. 0.67	0.2943	GSE57805	rat
37- glioblastoma (GBM) [brain] cisplatin 4128	Treatment => cisplatin vs. DMSO	0.4672	GSE97460	human
4- malignant pleural mesothelioma [mesothelium] cisplatin 6824	Treatment: Treat Time [hours] => 8 -> cisplatin vs. none	0.1641	GSE22445	human
5- malignant pleural mesothelioma [mesothelium] cisplatin; piroxicam 6825	Treatment: Treat Time [hours] => 8 -> cisplatin; piroxicam vs. none	0.1488	GSE22445	human

**Table 4 genes-14-02073-t004:** microRNAs and a siRNA interacting with the cisplatin network in gastric adenocarcinoma.

Symbol	Family
CAS3/SS3	biologic drug
miR-1195 (miRNAs w/seed GAGUUCG)	mature microRNA
mir-124	microRNA
mir-338	microRNA
mir-379	microRNA
mir-434	microRNA
mir-515	microRNA
mir-622	microRNA
MIR124	group
MIR499B	microRNA

**Table 5 genes-14-02073-t005:** Regulatory networks of arsenic-trioxide-treated liver carcinoma.

ID	Diseases & Functions	Known Regulator-Disease/Function Relationship
1	Cell death of carcinoma cell lines, cell proliferation of adenocarcinoma cell lines	11% (2/18)
2	Death of embryo, homologous recombination of DNA	0% (0/18)
3	Cell cycle progression, cell proliferation of tumor cell lines	65% (13/20)
4	Cell death of breast cancer cell lines, cell survival	0% (0/6)
5	DNA recombination, formation of γ H2AX nuclear focus, incidence of lymphoma	0% (0/15)
6	Chromosomal aberration, DNA damage, homologous recombination of DNA, T-cell non-Hodgkin lymphoma	13% (1/8)
7	Formation of γ H2AX nuclear focus	13% (1/8)
8	Chromosomal aberration, DNA damage, T-cell non-Hodgkin lymphoma	11% (1/9)
9	Death of embryo	0% (0/5)
10	Chromosomal aberration, DNA damage	13% (1/8)

**Table 6 genes-14-02073-t006:** microRNAs that have direct RNA–RNA interactions with a causal network of arsenic.

microRNAs
mir-101
mir-103
miR-125b-2-3p (and other miRNAs w/seed CAAGUCA)
mir-145
mir-22
mir-338
mir-379
mir-515
mir-622
MIR499B

## Data Availability

The datasets analyzed for this study can be found in the Gene Expression Omnibus (GEO) GSE81267 (https://www.ncbi.nlm.nih.gov/geo/query/acc.cgi?acc=GSE81267, accessed on 13 November 2023) and GSE6970 (https://www.ncbi.nlm.nih.gov/geo/query/acc.cgi?acc=GSE6907, accessed on 13 November 2023), and The Cancer Genome Atlas (TCGA) of the cBioPortal for Cancer Genomics database at the National Cancer Institute (NCI) Genomic Data Commons (GDC) data portal (https://portal.gdc.cancer.gov/, accessed on 13 November 2023).

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
