# Peer review of "Molecular Networks of Platinum Drugs and Their Interaction with microRNAs in Cancer"

_genes, 2023, doi:10.3390/genes14112073_

Round 1
Reviewer 1 Report
Comments and Suggestions for Authors
I thank the authors for submitting this manuscript entitled "Molecular networks of platinum drugs and their interaction with microRNAs in cancer" by Tanabe et al. The topic covered in the manuscript is potentially interesting from a strictly clinical point of view, offering useful insights to an oncology readership.
However, I feel that the experimental design is too broad, focusing on very different tumours (gastric adenocarcinoma, lung adenocarcinoma, B-cell lymphoma). This risks being confusing, mixing up different pathological realities, and potentially being treated differently.
I think the experimental design, as it was conceived, is a bit confusing, perhaps it would be better to rethink it on a certain type of tumour. In my opinion, I think creating biological interaction models with platinum-based chemotherapy as the only common denominator is not very scientific.
Comments on the Quality of English LanguageI would suggest a minor revision of English.
Author Response
Thank you for your very thoughtful and important comments. The study aims to investigate the precise mechanism of drug resistance in cancer in terms of metallodrugs including cisplatin. The manuscript has been revised to emphasize the importance of examining the molecular networks of anti-cancer drugs to reveal the drug resistance mechanism concerning epithelial-mesenchymal transition. The wide variety of anti-cancer drugs should be considered in terms of drug resistance, although platinum-based chemotherapy is the main focus of this paper. The strength and limitation of the study has now been added in the revised manuscript as follows:
(Lines 336-347): The main mechanism of drug resistance is a DNA-platinum drug complex, which induces a DNA repair system [7]. The limitation of this study is the lack of elucidating the exact mechanism regarding the correlation between platinum complexes and EMT. Platinum drug treatment could co-activate DNA repair mechanisms, cell cycle, and EMT signaling pathways. It is also possible that some EMT regulation is involved during platinum drug treatment. Clarifying these possibilities would be a future challenge.
In this study, we investigated the altered molecular networks in metallodrugs including platinum drugs and arsenic trioxide, and their relation to miRNAs, which will contribute to revealing the precise mechanism of drug resistance and possible combination therapy in clinical application. The identification of target miRNAs in molecular networks would lead to the development of cancer therapy and strategies for precision medicines in practice.
Reviewer 2 Report
Comments and Suggestions for Authors
The presentation of arsenic trioxide together with platinum complexes in this manuscript needs more explanation because it is no platinum metal at all, and just a metalloid. Some words about the safety of arsenic trioxide might be necessary because some readers might question its toxicity profile and suitability as a human drug.
Lines 105, 109, and 255: Better use the more common ´´DLBCL´´ instead of ´´DLBC´´.
The qualities of the figures are not good in my pdf copy of the manuscript, the letters and scripts are often too small and a little unclear. Maybe the authors can check and modify them if possible or necessary.
Table 3: The numbers in the ´´Analysis Name´´ column need to be explained. Is there a special reason why ´´cisplatin´´ is written in italics?
Discussion: In some parts of the discussion I miss the correlation with platinum complexes. I suggest to modify this section accordingly.
Reference 28: Please abbreviate the journal name.
Author Response
The presentation of arsenic trioxide together with platinum complexes in this manuscript needs more explanation because it is no platinum metal at all, and just a metalloid. Some words about the safety of arsenic trioxide might be necessary because some readers might question its toxicity profile and suitability as a human drug.
Thank you very much for your important comments. The explanation of the safety of arsenic trioxide has been added as follows:
(Lines 48-52): Arsenic trioxide has a long history of medicinal use for thousands of years, despite its toxic effects as a poison [10,11]. Arsenic trioxide is effective for patients with recurrent acute promyelocytic leukemia [12]. An increasing number of patients have been treated with arsenic trioxide, however, the precise mechanism of action is not fully understood [10,11].
<References>
- Au, W.-Y.; Kwong, Y.-L. Arsenic trioxide: safety issues and their management. Acta Pharmacologica Sinica 2008, 29, 296-304, doi:10.1111/j.1745-7254.2008.00771.x.
- Paul, N.P.; Galván, A.E.; Yoshinaga-Sakurai, K.; Rosen, B.P.; Yoshinaga, M. Arsenic in medicine: past, present and future. BioMetals 2023, 36, 283-301, doi:10.1007/s10534-022-00371-y.
- Lazo, G.; Kantarjian, H.; Estey, E.; Thomas, D.; O'Brien, S.; Cortes, J. Use of arsenic trioxide (As2O3) in the treatment of patients with acute promyelocytic leukemia: the M. D. Anderson experience. Cancer 2003, 97, 2218-2224, doi:10.1002/cncr.11314.
Lines 105, 109, and 255: Better use the more common ´´DLBCL´´ instead of ´´DLBC´´.
DLBC was changed into DLBCL (Lines 136, 140, 289).
The qualities of the figures are not good in my pdf copy of the manuscript, the letters and scripts are often too small and a little unclear. Maybe the authors can check and modify them if possible or necessary.
The figures have been modified to be as clear as possible. Figure 1d has been made larger and Figure 2a has been made clearer by analyzing the data in Tableau software.
The description on 2.3. Data Visualization was added as follows:
(Lines 120-122): 2.3. Data Visualization
The results of network analyses of causal networks of cisplatin treatment were visualized with Tableau software (https://www.tableau.com (accessed on 7 November 2023)).
Table 3: The numbers in the ´´Analysis Name´´ column need to be explained. Is there a special reason why ´´cisplatin´´ is written in italics?
The numbers in the “Analysis Name” column in Table 3 is the ID name of the Ingenuity Pathway Analysis database (As of 2022). The reason why “cisplatin” is written in italics is because the term “cisplatin” was searched for treatment in the database. The font of “cisplatin” was changed to the normal font in Table 3. The order of analysis names has been changed in the number according to Figure 2a. The descriptions have been added as follows:
(Lines 174-176): The numbers in the “Analysis Name” column in Table 3 are the ID name of the Ingenuity Pathway Analysis database (As of 2022).
Discussion: In some parts of the discussion I miss the correlation with platinum complexes. I suggest to modify this section accordingly.
Thank you for the nice suggestions. The correlation with platinum complexes has been added in Discussion as follows:
(Lines 336-347): The main mechanism of drug resistance is a DNA-platinum drug complex, which induces a DNA repair system [7]. The limitation of this study is the lack of elucidating the exact mechanism regarding the correlation between platinum complexes and EMT. Platinum drug treatment could co-activate DNA repair mechanisms, cell cycle, and EMT signaling pathways. It is also possible that some EMT regulation is involved during platinum drug treatment. Clarifying these possibilities would be a future challenge.
In this study, we investigated the altered molecular networks in metallodrugs including platinum drugs and arsenic trioxide, and their relation to miRNAs, which will contribute to revealing the precise mechanism of drug resistance and possible combination therapy in clinical application. The identification of target miRNAs in molecular networks would lead to the development of cancer therapy and strategies for precision medicines in practice.
Reference 28: Please abbreviate the journal name.
The journal name was abbreviated for Reference 38 (previously 28).
Reviewer 3 Report
Comments and Suggestions for Authors
First of all, I would like to thank you for inviting me to review the manuscript entitled: 'Molecular networks of platinum drugs and their interaction with microRNAs in cancer’. The aim of the study was to investigates the relationship between metallodrugs and EMT and tumor microenvironment pathways. Additionally, the molecular networks of carboplatin, cisplatin, and arsenic trioxide were investigated in the study. The general conclusion demonstrates the importance of the molecular networks of anti-cancer drugs and tumor microenvironment in the treatment of cancer resistant to anti-cancer drugs. Thus, I think that some major issues should be addressed:
1. Authors should extend introduction, since there is a lack of information related to side effects of those drugs, explain the meaning of EMT in more detail.
2. I suggest to add some mechanisms of cisplatin resistance from following manuscripts:
1) Chen SH, Chang JY. New Insights into Mechanisms of Cisplatin Resistance: From Tumor Cell to Microenvironment. Int J Mol Sci. 2019 Aug 24;20(17):4136. doi: 10.3390/ijms20174136.
2) Lugones, Y.; Loren, P.; Salazar, L.A. Cisplatin Resistance: Genetic and Epigenetic Factors Involved. Biomolecules 2022, 12, 1365. https://doi.org/10.3390/biom12101365.
3) Wang L, Zhao X, Fu J, Xu W, Yuan J. The Role of Tumour Metabolism in Cisplatin Resistance. Front Mol Biosci. 2021 Jun 23;8:691795. doi: 10.3389/fmolb.2021.691795.
4) Makovec, Tomaz. "Cisplatin and beyond: molecular mechanisms of action and drug resistance development in cancer chemotherapy" Radiology and Oncology, vol.53, no.2, 2019, pp.148-158. https://doi.org/10.2478/raon-2019-0018.
3. Also, in discussion section please add paragraph related to clinical and practical aspects of the study.
4. How we can applicate your results into practice?, why your work is valuable in the field?
Minor points:
1. Minor modification of the grammar and punctuation is required.
2. Please improve the quality of all figures.
3. Also, the symbols could be little bit bigger in all figures.
4. Please provide strength and limitation of the study.
General: interesting, well-conducted work.
Minor modification of the grammar and punctuation is required.
Author Response
First of all, I would like to thank you for inviting me to review the manuscript entitled: 'Molecular networks of platinum drugs and their interaction with microRNAs in cancer’. The aim of the study was to investigates the relationship between metallodrugs and EMT and tumor microenvironment pathways. Additionally, the molecular networks of carboplatin, cisplatin, and arsenic trioxide were investigated in the study. The general conclusion demonstrates the importance of the molecular networks of anti-cancer drugs and tumor microenvironment in the treatment of cancer resistant to anti-cancer drugs. Thus, I think that some major issues should be addressed:
- Authors should extend introduction, since there is a lack of information related to side effects of those drugs, explain the meaning of EMT in more detail.
Thank you for your valuable comments. The descriptions of the side effects of drugs and the meaning of EMT have been added in the introduction:
(Lines 56-59): Cisplatin treatment causes several toxic side effects including hepatotoxicity, cardiotoxicity, and nephrotoxicity, which may be based on the nature of cisplatin interacting with DNA and forming covalent adducts with purine DNA bases [6].
(Lines 86-94): It has been reported that miRNAs are involved in cisplatin response by regulating EMT [24]. EMT, a cellular phenomenon, where cell types change from epithelial to mesenchymal type, is involved in the malignancy of cancers including proliferation, invasion, and metastasis [25]. The EMT phenotype is regulated by miRNAs and plays important roles in drug resistance which is implicated with cancer stemness [25]. Tumor-derived exosomes containing several miRNAs promote EMT and cancer cell invasion [26]. Although EMT is involved in several signaling pathways such as phosphatase and tensin homolog (PTEN)/phosphatidylinositol 3-kinase (PI3K), Wnt/b-catenin, and TGF-b pathways, the precise correlation among EMT, exosomes and drug resistance is unknown [26].
<References>
24. Tolue Ghasaban, F.; Maharati, A.; Zangouei, A.S.; Zangooie, A.; Moghbeli, M. MicroRNAs as the pivotal regulators of cisplatin resistance in head and neck cancers. Cancer Cell Int 2023, 23, 170, doi:10.1186/s12935-023-03010-9.
25. Pan, G.; Liu, Y.; Shang, L.; Zhou, F.; Yang, S. EMT-associated microRNAs and their roles in cancer stemness and drug resistance. Cancer Commun (Lond) 2021, 41, 199-217, doi:10.1002/cac2.12138.
26. Mashouri, L.; Yousefi, H.; Aref, A.R.; Ahadi, A.M.; Molaei, F.; Alahari, S.K. Exosomes: composition, biogenesis, and mechanisms in cancer metastasis and drug resistance. Mol Cancer 2019, 18, 75, doi:10.1186/s12943-019-0991-5.
2. I suggest to add some mechanisms of cisplatin resistance from following manuscripts:
1) Chen SH, Chang JY. New Insights into Mechanisms of Cisplatin Resistance: From Tumor Cell to Microenvironment. Int J Mol Sci. 2019 Aug 24;20(17):4136. doi: 10.3390/ijms20174136.
2) Lugones, Y.; Loren, P.; Salazar, L.A. Cisplatin Resistance: Genetic and Epigenetic Factors Involved. Biomolecules 2022, 12, 1365. https://doi.org/10.3390/biom12101365.
3) Wang L, Zhao X, Fu J, Xu W, Yuan J. The Role of Tumour Metabolism in Cisplatin Resistance. Front Mol Biosci. 2021 Jun 23;8:691795. doi: 10.3389/fmolb.2021.691795.
4) Makovec, Tomaz. "Cisplatin and beyond: molecular mechanisms of action and drug resistance development in cancer chemotherapy" Radiology and Oncology, vol.53, no.2, 2019, pp.148-158. https://doi.org/10.2478/raon-2019-0018.
The mechanisms of cisplatin resistance have been added in the introduction with citation of the references you suggested as follows:
(Lines 54-69): The principal mechanism of cisplatin-induced cytotoxicity is the formation of platinum-DNA adducts, for which cellular reactions attenuating the DNA damage are key factors in resistance to cisplatin [13]. Cisplatin treatment causes several toxic side effects including hepatotoxicity, cardiotoxicity, and nephrotoxicity, which may be based on the nature of cisplatin interacting with DNA and forming covalent adducts with purine DNA bases [6]. Mechanisms of cisplatin resistance include pre-target resistance relating to the reduction of cisplatin uptake, the increase of cisplatin effusion to the extracellular space, or the increased sequestration of cisplatin by glutathione, on-target resistance involving molecular damage caused by DNA binding of cisplatin, post-target resistance including signaling pathways leading to cell death triggered by adducts, and off-target resistance relating to alterations in signaling pathways that interfere with cisplatin-induced proapoptotic events [14,15]. Furthermore, epigenetic mechanisms regulating gene expression in non-coding RNA basis and metabolic processes including glycometabolism and amino acid and lipid metabolism are associated with resistance to cisplatin treatment [14,15]. In terms of uptake and efflux of platinum drugs, polymorphisms of transporters influence the platinum drugs [16].
- Also, in discussion section please add paragraph related to clinical and practical aspects of the study.
A paragraph related to clinical and practical aspects of the study has been added:
(Lines 336-341): The main mechanism of drug resistance is a DNA-platinum drug complex, which induces a DNA repair system [7]. The limitation of this study is the lack of elucidating the exact mechanism regarding the correlation between platinum complexes and EMT. Platinum drug treatment could co-activate DNA repair mechanisms, cell cycle, and EMT signaling pathways. It is also possible that some EMT regulation is involved during platinum drug treatment. Clarifying these possibilities would be a future challenge.
- How we can applicate your results into practice? why your work is valuable in the field?
A paragraph on the application in practice has been added to Discussion:
(Lines 342-347): In this study, we investigated the altered molecular networks in metallodrugs including platinum drugs and arsenic trioxide, and their relation to miRNAs, which will contribute to revealing the precise mechanism of drug resistance and possible combination therapy in clinical application. The identification of target miRNAs in molecular networks would lead to the development of cancer therapy and strategies for precision medicines in practice.
Minor points:
1. Minor modification of the grammar and punctuation is required.
The grammar and punctuation have been checked.
2. Please improve the quality of all figures.
The quality of the figures has been improved.
3. Also, the symbols could be little bit bigger in all figures.
The symbols of the figure have been made bigger as possible.
4. Please provide strength and limitation of the study.
Strengths and limitations of the study have been added to Discussion:
(Lines 336-347): The main mechanism of drug resistance is a DNA-platinum drug complex, which induces a DNA repair system [7]. The limitation of this study is the lack of elucidating the exact mechanism regarding the correlation between platinum complexes and EMT. Platinum drug treatment could co-activate DNA repair mechanisms, cell cycle, and EMT signaling pathways. It is also possible that some EMT regulation is involved during platinum drug treatment. Clarifying these possibilities would be a future challenge.
In this study, we investigated the altered molecular networks in metallodrugs including platinum drugs and arsenic trioxide, and their relation to miRNAs, which will contribute to revealing the precise mechanism of drug resistance and possible combination therapy in clinical application. The identification of target miRNAs in molecular networks would lead to the development of cancer therapy and strategies for precision medicines in practice.
General: interesting, well-conducted work.
Thank you very much again for your kind comments to improve the manuscript. We greatly appreciate your deep insight and time for reviewing.
Round 2
Reviewer 1 Report
Comments and Suggestions for Authors
I thank the authors, S. Tanabe et al. for the extensive revision of the manuscript entitled ” Molecular networks of platinum drugs and their interaction with microRNAs in cancer”, following the suggestions.
The revision process has improved the manuscript, which is enough for me.
Reviewer 3 Report
Comments and Suggestions for Authors
I suggest to publish the manuscript in current form.